# Exposure Assessment to Deoxynivalenol of Children over 3 Years Deriving from the Consumption of Processed Wheat-Based Products Produced from a Dedicated Flour

**DOI:** 10.3390/toxins15100615

**Published:** 2023-10-16

**Authors:** Carlo Brera, Caterina De Santis, Stefania Marzona, Emanuela Gregori, Sabrina Santa Prisco, Maurizio Monti, Gabriele Chilosi, Anna Pantanali

**Affiliations:** 1Independent Researcher, 00175 Rome, Italy; caterina.desantis92@gmail.com (C.D.S.); sabrinasanta.prisco@gmail.com (S.S.P.); 2Molino Moras, 33050 Trivignano Udinese, Italy; stefania.marzona@gmail.com (S.M.); anna@molinomoras.it (A.P.); 3Italian National Institute of Health, 00161 Rome, Italy; emanuela.gregori@iss.it; 4Millers’ Mastery, 40020 Sassoleone, Italy; maurizio.monti@gmail.com; 5Agrifood and Forest Systems (DIBAF), Department of Innovation in Biology, University of Tuscia, 01100 Viterbo, Italy; chilosi@unitus.it

**Keywords:** deoxynivalenol, prevention measures, exposure assessment, public health, vulnerable groups, wheat flour, HACCP, bakery products

## Abstract

Wheat-based products are largely consumed by children worldwide. Deoxynivalenol (DON) is known for its acute and chronic toxicity and is the most common contaminant of cereal grains. Since no legal limits are set for DON in wheat-based products and specific foods intended for children over 3 years on the market, a high risk of overexposure to this contaminant may emerge. The main objective of the study, conducted in 2018–2019, was to produce a wheat flour intended for children over three years, characterized by a high level of safety in terms of DON content, to be used to produce wheat-derived products. The dedicated flour was produced by adopting tailored procedures like the selection of wheat suppliers, the predetermination of the safe contamination of DON in the final products, and the evaluation of the transfer rate from the wheat flour to derived products (bread, breadsticks, biscuits, plumcake, and focaccia). The results showed that the daily exposure of children was considered to be safe, in a range between 7% (biscuits) and 67% (bread) of DON tolerable daily intake (TDI) and that only by producing a flour characterized by DON levels much lower than those in force, can “safe” products be marketed.

## 1. Introduction

Food safety entails health and hygiene requirements that must characterize any food product put on the market. To ensure the health of the end consumer, these requirements must stem from the adoption of good agricultural practices (GAP) and compliance with the legal limits of residues from pesticide or veterinary treatments and industrial, environmental, and agricultural contaminants. This includes implementing proper storage conditions of the raw materials and the final products, preventing alterations in the production process, and maintaining correct transport and marketing conditions. The actions mentioned above should be put in place in order to meet Article 14, paragraph 1, of Regulation CE/178/2002 [1] on food safety requirements, where it is clearly reported that “Food shall not be placed on the market if it is unsafe”.

However, the Regulation does not indicate what is specifically meant by “unsafe” and how to avoid placing such products on the market. Paragraph 2 of the same article merely describes an unsafe product as harmful to health or not suitable for human consumption, identifying in paragraph 3 some requirements related to the provision of information on the label, the hygienic–sanitary status of the product, the long-term toxicity, and particular attention towards the more vulnerable consumer classes. As is known, food business operators (FBOs) are responsible for specific obligations of action and intervention in case of risks to public health due to the introduction of unsafe food products. Nonetheless, Article 9 of the relatively recent EU/2017/625 Regulation [2] strengthens the concept relating to the safe use of food products, under which the competent authorities regularly carry out official controls in the different food chains based on risk analysis and with appropriate frequency. Therefore, asserting that the absence of risk, or at least the presence of a minimum risk, should represent a safety pre-requisite characterizing every food product placed on the market.

In this perspective, both FBOs and risk managers are asked to adopt prevention measures and set suitable legal maximum limits, respectively. In addition, according to the precautionary principle (Article 7 of Regulation EC/178/2002), it is also imperative to promote production systems aimed at reducing the presence of contaminants to the lowest possible levels (sustainable by the supply chain or by the company), in order to minimize the risk associated with accumulation and toxicity in the long term.

It is noteworthy to highlight that even if food products are marketed at contamination levels lower than legal maximum levels, in some cases, a risk for vulnerable consumer groups such as children over 3 years may still be present.

In this context, the project “Research and development of an innovative flour for children over three years”, financed by the Friuli Venezia Giulia Region and Molino Moras Company, was launched in 2018 as a result of a collaboration between the Istituto Superiore di Sanità (ISS, Rome, Italy) and Molino Moras Company (Trivignano Udinese, Udine, Italy), and by a research collaboration agreement signed between the ISS and the Department for Innovation in Biological, Agro-food, and Forest systems (DIBAF), of the “Università degli studi della Tuscia” in Viterbo (Italy).

The overall aim of the project was to develop operational procedures within the soft wheat sector to produce a “dedicated” type of flour that is, from a health perspective, suitable, for the production of soft wheat-derived products consumed by vulnerable consumer groups like children. The “vulnerable” consumer category of this project was the population group of children aged between 3 and 9.9 years, defined as “Other Children” in the EFSA official consumption database [3]).

Moreover, the “dedicated” flour, the object of this study, refers to a flour characterized by safe levels of pathogens and contaminants, including deoxynivalenol (DON), a mycotoxin quite commonly found in wheat.

The rationale supporting the project was based on the following points:the marketing of food products at contamination levels lower than MLs could not correspond to a condition of safety for some vulnerable consumer groups, like children;there is no specific legal limit set for any of the contaminants and food products for the consumer group of children aged 3.0 to 9.9, since, for this category, we must consider that legal limits for the same as products consumed by adults;the exposure of children aged 3.0 to 9.9, corresponding to the daily consumption of food products containing specific contaminants, at levels of contamination even lower than legal limits, can be higher than the reference toxicological threshold set for the contaminant. Therefore, it is noteworthy to individuate a level of contamination for DON in flour that is lower than the current legal limit, consistent with safe contamination levels in some derived wheat-based products daily consumed by children aged 3.0 to 9.9.

From the above, the exposure of children aged 3.0 to 9.9 can pose a high risk, especially considering this age group’s high daily consumption of wheat flour-based products, occurring in Italy and worldwide. Hence, it is noteworthy to consider that, in some cases, the maximum level (ML) may not correspond to an adequate level of safe consumption for some vulnerable consumer groups, being a net difference in the concept behind the ML and the safe level (SL) for which a toxicological reference threshold is set.

Indeed, while ML is based on the need to minimize the presence of contaminants in food as much as possible [4], SL corresponds to a level of hazard and risk in food at or below that which the food is considered safe, according to its intended use [5].

Within this net distinction, ML and SL could not coincide or be characterized by different starting assumptions. Therefore, the study was conducted taking into account the current maximum tolerable limits of DON as set in the EU legislation, the toxicological threshold of reference for DON set by European and international organizations, the definition of “child” as reported by UNICEF (Convention on the Rights of Childhood and Adolescence, UNICEF 1989) [6], and the EFSA definition of the category “other children”.

As far as the current maximum tolerable levels of DON in food products, the European Commission introduced with the EC/1881/2006 Regulation [7] the maximum legal levels for DON in some food products, including wheat (1250 µg/kg), flour (750 µg/kg), cereal-based products (500 µg/kg) and cereal-based products intended for infants (200 µg/kg). It must be emphasized that, very recently, Regulation EC/1881/2006 was repealed by Regulation (EU) 2023/915 [8], with no substantial difference regarding DON maximum levels in wheat-based products from the previous ones. In addition, due to the paucity of available data, no maximum limit was set for the DON metabolites found in plants and food products, i.e., 3- and 15 acetyl-DON and DON-3-glucoside. For this reason, this study considered the exposure assessment only of the parent compound DON and not its metabolites.

As is known, the reference toxicological threshold for non-carcinogenic substances is represented by the NOAEL (no-observed-adverse-effect-level) that, multiplied by a safety factor, leads to a health-based guidance value (HBGV) that, in the case of DON for humans, is the tolerable daily intake (TDI). This HBGV for DON was set at 1000 ng/kg bw/day and recently extended to the group of the above-mentioned three major metabolites commonly co-occurring with the parent toxin [9].

In 2011, the Joint Expert Committee on Food Additives (JECFA) derived a group acute reference dose (ARfD) of 8 μg/kg bw for DON and its acetylated derivatives using the lowest limit on the benchmark dose for a 10% response (BMDL10) of 0.21 mg/kg bw per day for emesis in pigs [10].

Children are commonly classified into three subgroups with different characteristics, i.e., infants from 0 to 1 year, toddlers aged between 1 and 3 years, and pre-schoolers over 3 (Central Disease for Control and Prevention, CDC, available at https://www.cdc.gov/ncbddd/childdevelopment/positiveparenting/preschoolers.html, page last reviewed: 22 February 2021). Consistently with this distinction, the food products available on the market are produced differently depending on whether they are intended for infants or toddlers (baby foods) or children over 3 years. Baby foods are subjected to specific and much more strict legislative provisions about the presence of contaminants, with maximum limits set much lower than the ones applied to children over three years. From the above, it is clear that no distinction in maximum limits of a contaminant in a food product between a child of a 3-year-old and all other consumer groups exists. This condition can lead, for the same combination of contaminant food products and similar consumption rates among the different consumer groups, to an over-exposure in the case of children and a safe exposure in the case of all the other groups, i.e., adolescents, adults, and the elderly.

Among the contaminants affecting most crops, including soft and durum wheat, mycotoxins play a crucial role from an economical, ethical, environmental, and health point of view. Mycotoxins are natural toxins produced by the secondary metabolism of some fungal species, mainly belonging to the genera *Aspergillus*, *Penicillium*, and *Fusarium*. More than 500 mycotoxins are currently known, produced by a broad spectrum of fungal species, which have remarkably differentiated chemical structures [11].

As is known, wheat (*Triticum aestivum* and *Triticum durum*) is one of the crops subject to numerous plant diseases of fungal origin, of which Fusarium head blight (FHB) is one of the most widespread and harmful. Over 17 different fungal species, belonging to the genus *Fusarium* and *Microdochium nivale*, have been associated with FHB, and the most important species for diffusion and pathogenicity are *F. graminearum* and some fungal species belonging mainly to the genera *Aspergillus*, *Penicillium*, and other *Fusarium* (*F. culmorum*, *F. avenaceum*, and *F. poae*). The main factors responsible for the growth of these fungi are biotic, such as pest attack and fungal strain, and abiotic, such as environmental stresses to which the plant has been subjected (e.g., conditions of extreme aridity of the field, temperature fluctuations, and lack of balanced absorption of nutrients) [12,13].

DON is the most common contaminant of cereal grains, particularly soft and durum wheat and their derived products. Its presence in food and feed is found in over 90% of samples and is often an indicator of contamination from other mycotoxins. DON is also known as “vomitoxin” due to its strong emetic effects in humans and animals (pigs in particular) [9]. In addition to emesis, the toxic effects of DON include anorexia, impaired intestinal and immune functions, nausea, reduced absorption of nutrients, and high susceptibility to infections and chronic diseases [9].

In the review by EFSA [9], several outputs of exposure assessment studies in humans have been reported. In the majority of cases, young children were the most exposed consumer category. In 2011, JECFA [14] performed a study on the exposure assessment of DON through wheat and derived product consumption. The study, based on GEMS/Food consumption cluster diets, revealed that almost all countries showed exposure rates of concern, ranging from 0.11 μg/kg bw per day (cluster L) to 9.13 μg/kg bw per day (cluster M). The same study assessed acute exposure to bread as the most representative food, considering its higher daily consumption. Based on the concentration of 1 mg DON/kg, food, the estimated acute dietary exposure was 9 µg/kg bw per day, exceeding the reference value of 8 µg/kg bw per day.

In a recent study, Gallardo et al. [15] assessed the exposure of the Spanish and Catalonian region population’s daily consumption of cereal-based foods. It showed that exposure to DON was low in the adult population, while children aged 3–9 years were heavy consumers, with higher exposure values up to 604 ng/kg body weight/day being reached, which was very close to the reference toxicological threshold. Bread and pasta were the main contributing foodstuffs to the global exposure to DON, posing a concern for the health of this category and forcing food control activities aimed at reducing the exposure.

In 2019, Chen et al. [16] published an interesting review on the exposure assessment calculated worldwide. It was concluded that for populations of children of different age groups, dietary DON exposure for 2–6-year-old children in Australia, children of all ages in China, and adolescents in Belgium were higher than the TDI; these outputs led to the hypothesis of the association between chronic gastrointestinal and immunomodulatory effects with wheat consumption in their diets. In addition, an exceeding of the acute reference dose was also observed in certain populations of China where large amounts of wheat are consumed daily.

More recently, a review by Wang et al. [17] on DON exposure derived from wheat consumption in different countries showed that most of the investigated countries presented exposure rates lower than the TDI, even if some susceptible populations like children showed dietary exposures higher than the TDI value in the highest percentile. It should also be noted that these results referred only to wheat consumption without considering the cumulative exposure from other sources of DON.

In 1993, The International Agency for Research on Cancer (IARC), concluded that “there is insufficient evidence on carcinogenicity of DON in animal experiments”, classifying DON in group 3 as a “non-carcinogenic agent for humans” [18]. DON shows low acute toxicity compared to other mycotoxins of the same class.

However, being present in a wide variety of products, it introduces a non-negligible level of risk for the consumer, both for the possible overall effects due to the sum effect and for those of long-term toxicity. Furthermore, DON is a rather heat-stable mycotoxin, presenting resistance to the temperatures typical of cooking and frying processes [19,20].

This chemical–physical property implies its detection in processed and cooked products [9].

## 2. Results and Discussion

### 2.1. Evaluation of the Transfer of the DON from the Raw Material to the Finished Products

To obtain information on the fate of deoxynivalenol as a result of the first (milling) and the second transformation process, three different wheat grains were milled to obtain corresponding flours, characterized by different levels of DON contamination indicated as “low” (172.3 µg/kg), “medium” (276.9 µg/kg), and “high” (332.2 µg/kg). All of these flours were both organic and conventional of EU and Italian origin, type 0, and were obtained from single and multi-varietal soft wheat grains.

The loss of DON due to the milling process is shown in Table 1 and ranged from 53.3% to 61.1%. Each of the three selected flours was then used to produce five processed products, such as bread, focaccia, breadstick, biscuit, and plumcake, according to specific recipes commonly used in domestic preparations. This process was repeated three times. Then, the abatement in the percent of the DON content from the flour to the processed products was calculated considering the initial level of contamination of the flour, the formulation used, and the process applied. In general, the average reduction percentage was not higher than 22% except in the case of biscuits and cakes, where the average percentage was around 10% (Table 2). These results reflect what has been reported by other authors in the past, as discussed below.

Generally, DON is highly thermal stable. However, in previous studies, discrepancies in the fate of DON after the baking process were observed. A review by Kushiro in 2008 [21] reported controversial results, with studies showing no reduction in DON concentration at the temperature of 170–350 °C after 30 min at 170 °C [22], while other studies, with more similar results to those obtained in the present study, showed a decrease of up to 35% in DON during baking biscuits [23]

In 2014, Vidal et al. [22] reviewed a dozen studies on the effect of bakery processing on DON contamination in wheat products (bread, cakes, and biscuits). As far as baking is concerned, most studies have reported a reduction in DON concentrations of up to 79%, with the degree of reduction affected by the baking time, temperature, and loaf size. It was also concluded that the loss of DON in finished products could also be attributed to dilution factors due to the presence of other ingredients, such as fat, sugar, and water, rather than by thermal degradation [9,24]. This controversial behavior could derive from food additives, such as ammonium carbonate contained in batter [23]. Our explanation for the slight differences observed in our experiments is oriented at identifying the temperature/cooking time ratio, which is product-specific, and the amount of flour considered in the recipes of the specific products as the main causes. A possible formation of DON isomers and/or metabolites during the whole process of milling and baking was also postulated [25].

In our study, to calculate the transfer of DON from the starting flours to the five different derived products, the following formula was used:(1)$$\frac{\frac{ng}{g}flour-\frac{ng}{g}flour\;in\;fp}{\frac{ng}{g}flour}\times 100$$
where ng/g flour is the contamination level of DON in the starting flour and ng/g flour in fp corresponds to the contamination of DON in the flour contained in the finished product (fp).

### 2.2. Exposure Assessment

Exposure assessment is an integral part of the risk assessment; it is the qualitative and/or quantitative assessment of the intake of a chemical, physical, or biological agent basically through three different routes: by diet, inhalation, and/or contact [26].

Exposure assessment generally combines contamination data with official consumption data and body weight. Regarding the quantitative assessment of exposure to mycotoxins, two approaches can be highlighted, deterministic and probabilistic, which differ in how the data used are available. The deterministic approach is based on the analysis of single average values (average consumption multiplied by average concentration). In contrast, the probabilistic approach expresses the variables of consumption, concentration, and body weights in the distribution [26]. In this study, the deterministic approach was used since the individual distribution of the consumption data and body weights were unavailable.

With the object of characterizing this “dedicated” flour, it was considered that an acceptable exposure assessment—obtained from the consumption of the processed products investigated in the project—should not be higher than 75% of TDI. According to the algorithm used for the exposure assessment, reported in the following paragraphs, the identification of the target DON contamination level in the flour was individuated not higher than 200 µg/kg, corresponding to the highest threshold beyond which a risk for the children’s health may occur.

As far as the deterministic calculation is concerned, the following algorithm was used:Exposure (ng/kg bw/day) = DON Contamination level (µg/kg) × Consumption data (g)/body weight (kg)

The exposure to DON by children aged 3–9.9 years, expressed as the average and 95th percentile exposure, was calculated for each of the five finished products, as produced using the flour at the “low” contamination level (172.3 µg/kg). The low value was chosen because of its closeness to the target contamination level in the flour that was individuated for the scope of the research. In the algorithm, the contamination level corresponds to the average level found in the processed products, and the mean and 95th percentile consumption data reported for children aged 3–9.9 years, belonging to the category “Consumers only”, was derived from the official Italian Consumption database [27]. The average body weight for this consumer category is derived from the same publication used for the consumption data, equal to 26.1 kg [27].

For four out of five processed products, the corresponding daily consumption data, as reported in the Italian official database, was used [27]. For breadsticks, no specific indication of daily consumption data was available, thus, the data closest to this type of food was used, i.e., the aggregate data “refined salted baked goods”. The results related to the average exposure rates, as calculated from the individual consumption of five finished products (bread, focaccia, breadsticks, biscuits, and plumcake), produced by flour at the lower level of contamination, and the cumulative daily exposure value, i.e., the exposure calculated considering the daily consumption of all five matrices, are presented in Table 3. With reference to the toxicological threshold, namely the tolerable daily intake (TDI) for DON (also including its metabolites), set by the EFSA at 1000 ng/kg bw/day [9]), we can observe that:considering the mean and 95th percentile exposure rates, corresponding to the consumption of each individual product, the consumption of none of these products led to an exposure that exceeded the TDI. More specifically, the mean exposure corresponding to the consumption of the single products ranged from 33% of TDI for bread to 7% for biscuits. Higher exposure rates, but always lower the 75% of TDI, were obtained in the case of strong consumers (95th percentile);considering the cumulative exposure, only the mean exposure guarantees a value below the TDI; vice versa, in the case of strong consumers, an excess of TDI of 117% occurs.

We should also consider, even if for purely indicative purposes, the calculation of the exposure deriving from a DON contamination in food products equal to the legal limit (500 µg/kg) for bread, focaccia, breadsticks, biscuits, and plumcake [7,8] (Table 4). For this case, the obtained results were very alarming. For bread, the mean exposure was 58% higher than the TDI and 344% higher at the 95th percentile of consumption. In addition, considering the cumulative mean exposure, the value was five times higher than the TDI, and twelve times in the case of the strong consumers. Even if this scenario is quite far from being close to the daily reality, it should be considered that daily Italian food habits include all these products, and for this reason, it is worth highlighting.

From the overall results obtained, it was, therefore, possible to draw useful information to achieve the goal of the project, that is, the identification of an “action level of DON” in flour which is consistent with safe exposure through the consumption by children over three years of the most commonly consumed cereal-based products.

As shown, it was possible to obtain finished products that fully comply with the predetermined objective, starting from the flour with a low level of 172.3 µg/kg. These levels, in fact, guarantee an exposure for children that can be considered safe for their health. Based on the contamination value mentioned, it was therefore identified, for flour, an action level of 150 µg/kg ± 15%, i.e., three times lower than the current maximum legal limit.

It is also necessary to stress that the production cycle of the “dedicated flour” must necessarily be characterized by a robust and not occasional process to guarantee the consumer the systematic use of a product characterized by certified health and hygiene high-quality standards and in full agreement with the reference toxicological parameters set by EFSA.

## 3. Conclusions

Keeping in mind the objective of the project, i.e., to put into practice what is reported by article 14 of Regulation EC/178/2002 [1], a flour characterized by a safe level of deoxynivalenol intended for the safe daily consumption of wheat-derived products by children over 3 years, was produced. This was met by the use of targeted production process strategies to obtain finished products characterized by high-quality standards in terms of hygienic and sanitary characteristics. In particular, for children over three years, ensuring compliance with health conditions to protect their health systematically would be required.

What should be clear is that even if an appropriate sanitary standard should characterize any food product, it is not sufficient to market products at a level of contamination slightly lower than the legal limit since, in specific cases, it is necessary to individuate specific action limits, much lower than the legal ones, consistent with the risks derived by the consumption of the processed foods. In other words, in some cases, the targeting of the legal limit could not correspond to a condition of the requested safety for specific food products and vulnerable consumer groups, such as children over three years.

Therefore, according to the principle of precaution, the production cycles should take into account even the long-term effects on children’s health related to the cumulative exposure to contaminants. This is possible only by the adoption of activities aimed at a systematic reduction in levels of contaminants such as, for example, DON. To this end, the adoption of preventive measures appears to be the most efficient activity to achieve this objective in a structured way, especially for food products with a high dietary daily consumption. Finally, it should be emphasized the importance of clearly communicating to consumers, transparently and truthfully, the sanitary quality that describes the product on the market in order to respect the consumers’ “right to know” and the corresponding “right to choose”.

Finally, it is to be noted that this “dedicated” flour represents the first product marketed in Italy to ensure safe derived products consumed by children over 3 years.

## 4. Materials and Methods

### 4.1. Sampling

The sampling of soft wheat grain and flours was carried out directly by the Molino Moras technician, following the official procedure described in Regulation (EC) no. 401/2006 [28]. Samples were stored under vacuum and at −20 °C, then ground and/or homogenized and sub-sampled as appropriate to form the laboratory sample. In order to guarantee a representative sampling, 3 kg wheat samples were ground in two steps: a first coarse milling by a Romer RAS Mill and a second step by using a Retsch ZM 200 Mill with a sieve regulated at 750 µm. The corresponding flour samples were randomly selected to form sub-samples of about 500 g, from which the test aliquots were taken for the successive analytical step. The samples of the processed products (bread, focaccia, breadstick, biscuit, and plumcake) were prepared at Molino Moras according to specific recipes commonly used by consumers. For each finished product, three productions were made. Each production was sampled, vacuum-packed, blast chilled, and stored at −20 °C at the time of the analysis.

### 4.2. Method of Analysis

All samples were analyzed by a validated HPLC method using a UV detector after clean-up using an immunoaffinity column (IAC). More specifically, 20.0 g of the matrix was extracted with water, filtered, and passed through an IAC (DONPREP^®^ R-Biopharm AG, Darmstadt, Germany). After washing the IAC with water, the cleaned-up extract was eluted by passing two aliquots of 750 µL of methanol and completing the elution by flowing an air volume equal to 2/3 times the volume of the cartridge. After a complete drying by a flow of nitrogen at 40 °C, the eluate was reconstituted with 1 mL of a water/methanol 90.5/9.5 solution and injected into the HPLC system.

The operating conditions for the quantitative analysis were:chromatographic column: RP-Symmetry^®^ C18, 3.5 µm, 4.6 mm × 150 mm, regulated at 40 °C;injection volume: 150 µL;mobile phase: water:methanol HPLC grade 85:15 *v*/*v*;flow: 1.0 mL/min;detector UV: λ = 220 nm.

The quantitative analysis was performed by interpolation with a previously built six-point calibration curve. No interfering compounds were found for any of the investigated food matrices.

### 4.3. Verification of the Validated Reference Method of Analysis

Following the EURACHEM approach [29], the method of analysis used for DON determination in all products aimed to achieve the recommended fit-for-purpose performance characteristics reported in Regulation EC/401/2006 (Table 5 and Table 6). The obtained results fully complied with the requirements reported in the Regulation, with recovery factors higher than 80% and standard deviation under repeatability conditions always lower than 15%.

Similar results were obtained for the processed products (Table 6), where each cluster subject to the validation activity consisted of a composite sample prepared by mixing equal quantities of the different products composing it. The grouping logic was made, taking into consideration the recipe and preparation.

### 4.4. Methodological Process Used in the Study

To achieve the above-described goal, a series of activities were carried out to identify those elements necessary to ensure that the pre-evaluated safety requirements were met. The activities carried out were: the identification of the criteria for the selection of suitable suppliers of the soft wheat (GAP in place at the level of the farmer, good manufacturing procedures in place, such as proper storage conditions, sanitation of sound construction warehouses and silos, and proper transport conditions, as well documentation in line with the provided declarations); set-up of a standardized, sustainable, and systematic monitoring plan of the suppliers of the soft wheat; the definition of quality standards of the raw material necessary to obtain the final product with the requested characteristics; the identification of a raw material processing method solely dedicated to the production of this new type of flour; the characterization of the “dedicated” flour aimed at evaluating the action limit for DON; proper cleaning conditions of the production line in place at the mill; proper management conditions to downgrade the flour without the requested characteristics; the evaluation of the DON transfer from grain to flour and from flour to the finished products (bread, focaccia, biscuits, breadsticks, plumcake); the verification of the performance characteristics of a consolidated validated reference analytical method (see Materials and Methods section); the individuation of the target exposure risk by children consuming the investigated products on a daily basis; and the risk characterization by comparing the results of the exposure assessment with the reference toxicological dose.

In more detail, suppliers of the raw material (soft wheat) were duly investigated by means of selection based on specific criteria, such as the geographical area of wheat cultivation, the recurrent climatic conditions in the geographical area of production, i.e., those regions of the world where a rainy climate close to the anthesis period is rare, and the constancy in providing wheat with low levels of DON. The final choice was taken after replicating analyses of grains as received by three suppliers. It was decided to select the one that met the above criteria, and the “dedicated” flour was obtained from conventional, monovarietal wheat grains of Austrian origin.

The production cycle of the flour was carried out at the historical Molino Moras mill, which has a production capacity of 27 quintals/h.

The technical specifications of the “dedicated” flour were the following:eighty percent reduced DON content;no plant protection products residues;no glyphosate.

## 5. Patents

Since 2019, Molino Moras has held the patent for the “dedicated” flour in accordance with ISO norm 22005:2008 and three other voluntary product certifications.

## Figures and Tables

**Table 1 toxins-15-00615-t001:** Loss rate of DON from wheat grains in the corresponding flours (μg/kg).

Product	DON µg/kg	Relative Standard Deviation RSDr % *	% Loss
Wheat grain_7	358.9	4.1	63.6
“Low” level flour	172.3	4.4	
Organic wheat 8_9	711.2	0.4	61.1
“Medium level” flour	276.9	2.5	
Organic wheat 8_9	711.2	0.4	53.3
“High level” flour	332.2	5.1	

* Under repeatability conditions.

**Table 2 toxins-15-00615-t002:** DON content in finished products and loss from starting flours to the finished products.

DON in the Starting Flour (μg/kg)	Finished Product	DON (μg/kg)	Loss, %
Low level flour	Bread	105.3	16.0
172.3	Focaccia	92.5	20.9
	Breadsticks	145.8	18.1
	Biscuits	66.0	13.7
	Plumcake	60.5	11.4
Medium level flour	Bread	164.2	17.3
276.9	Focaccia	187.3	17.7
	Breadsticks	225.2	21.9
	Biscuits	113.0	8.0
	Plumcake	98.5	9.1
High level flour	Bread	212.2	13.6
332.2	Focaccia	220.4	10.3
	Breadsticks	273.8	15.2
	Biscuits	136.5	12.9
	Plumcake	131.7	2.2

**Table 3 toxins-15-00615-t003:** The mean and 95th percentile exposure rates corresponding to the daily consumption of cereal-based products prepared by the flour at a low contamination level (172.3 μg/kg).

Matrix	DON (μg/kg)	Mean Consumption (Grams)	95th Percentile Consumption (Grams)	Body Weight (kg) [27]	Mean Exposure (ng/kg bw/Day)	95th Percentile Exposure (ng/kg bw/Day)
Bread	105.3	82.6	180	26.1	333.2	726.2
Focaccia	92.5	39.9	100		141.4	354.4
Breadsticks	145.8	17.1	38		95.5	212.3
Biscuits	66.0	27.2	69.3		68.8	175.2
Plumcake	60.5	40.2	101.4		93.2	235.0
Cumulative exposure					997.9	2169.4

**Table 4 toxins-15-00615-t004:** DON exposure calculation at a legal limit (DON = 500 µg/kg).

Matrix	DON (μg/kg)	Mean Consumption (Grams)	95th Percentile Consumption (Grams)	Body Weight (kg)	Mean Exposure (ng/kg bw/Day)	95th Percentile Exposure (ng/kg bw/Day)
Bread	500	82.6	180	26.1	1582.4	3448.3
Focaccia	500	39.9	100		765.4	1915.7
Breadsticks	500	17.1	38		327.6	728.0
Biscuits	500	27.2	69.3		521.1	1327.6
Plumcake	500	40.2	101.4		770.1	1942.5
Cumulative exposure					5683.9	12,376.4

**Table 5 toxins-15-00615-t005:** Method performance characteristics for DON determination in grain and flour.

DON Level (µg/kg)	Low Level (Incurred)	50.0 Fortified	200.0 Fortified
Number of replicates	9	8	8
Mean, µg/kg	19.5	41.9	187
SDr *, µg/kg	3.6	1	5.9
Relative standard deviation (RSDr) ** (%)	12	2.4	3
Relative standard deviation (RSDR) *** (%)	19	15	7
Expanded uncertainty (µg/kg) (k = 2; 95% confidence interval)	6.5	12.5	50
Recovery (%)		84	84.3
Limit of quantification (µg/kg)	19.5		

* Standard deviation, calculated from results generated under repeatability conditions. ** Under repeatability conditions. *** Under intermediate repeatability conditions.

**Table 6 toxins-15-00615-t006:** Method performance characteristics for DON determination in wheat-based processed products.

Product	Bread, Focaccia, Breadsticks Cluster	Biscuits, Plumcake Cluster
Number of samples	6	6
Recovery (%), DON = 100 µg/kg	82	97
Relative standard deviation (RSDr) *, (%)	5	13
Recovery (%), DON = 200 µg/kg	85	88
Relative standard deviation (RSDr) *, (%)	4	5

* Under repeatability conditions.

## Data Availability

Data is unavailable due to privacy restrictions.

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
