# Peer review of "Exposure Assessment to Deoxynivalenol of Children over 3 Years Deriving from the Consumption of Processed Wheat-Based Products Produced from a Dedicated Flour"

_toxins, 2023, doi:10.3390/toxins15100615_

Round 1

Reviewer 1 Report

In general, the research is good, but it cannot be published without a comprehensive amendment. I have some observations: for example, the abstract must contain information about the subject before the objectives, and it does not mention the period of conducting the research. As for the introduction, it is very long and contains a lot of information about the research project, so there is no need to mention it, as if the researcher is writing a report on his research project. Also, there is confusion between the different parts of the research, for example, some tables are present in the research materials and methods, and some tables are present in the results. References need to be updated, so the latest reference is in 2017, and the research does not contain a reference in the period (2020-2023).

Reviewer 2 Report

Manuscript titled: Exposure assessment to deoxynivalenol by children over 3 years

deriving from the consumption of processed wheat-based products produced from

an innovative flour

Journal: toxins

Dear Editors,

The review manuscript entitled " Exposure assessment to deoxynivalenol by children over 3 years deriving from the consumption of processed wheat-based products produced from an innovative flour" for toxins has been completed. The manuscript entitled " Exposure assessment to deoxynivalenol by children over 3 years deriving from the consumption of processed wheat-based products produced from an innovative flour " has demonstrates that only by producing a flour characterized by DON levels much lower than those in force can safe products be achieved. In my opinion, the manuscript must be major revised for journal. I declare that I have no competing interests. More detailed comments are as follows:

The revised position:

1.     As a development of a new type of human flour, I believe that a standard for evaluating children's tolerance should be described in detail in the Materials and Methods section, along with the process and ethics of complying with EU ethical testing standards, as well as the ethical test number for informed consent of participants.

2.     All references cited in the article should strictly follow the journal's author's guide.

3.     Please clearly state the four different groups in the abstract.

4.     The manuscript has a lot of all kinds of errors, missing or wrong spaces, awkward wordings. It's hard to read. I have to edit it by two native English speakers and then submit it to other magazines.

5.     All of references should be carefully check out. Some important reference must be added.

6.     From the perspective of scientific discussion, in the discussion part, it is suggested to increase the discussion of persimmon polysaccharide synthesis on animals or cells after exposure of different stimulus sources to animals or cells, so that the discussion part of the article is more complete, and also reveals the intervention effect of persimmon polysaccharide synthesis on intestinal inflammation from different animal models and levels. Through PubMed search, we suggest adding these important literatures to make the discussion part of the article more complete. These important documents including doi:10.1016/j.fct.2023.113863; doi:10.2174/1389200221666200302124102; doi: 10.1080/10408398.2021.1895056;doi:10.1111/1541-4337.12545;doi: 10.1016/j.foodchem.2021.131978. You should support your discussion and results with these recent major publications.

Manuscript titled: Exposure assessment to deoxynivalenol by children over 3 years

deriving from the consumption of processed wheat-based products produced from

an innovative flour

Journal: toxins

Dear Editors,

The review manuscript entitled " Exposure assessment to deoxynivalenol by children over 3 years deriving from the consumption of processed wheat-based products produced from an innovative flour" for toxins has been completed. The manuscript entitled " Exposure assessment to deoxynivalenol by children over 3 years deriving from the consumption of processed wheat-based products produced from an innovative flour " has demonstrates that only by producing a flour characterized by DON levels much lower than those in force can safe products be achieved. In my opinion, the manuscript must be major revised for journal. I declare that I have no competing interests. More detailed comments are as follows:

The revised position:

1.     As a development of a new type of human flour, I believe that a standard for evaluating children's tolerance should be described in detail in the Materials and Methods section, along with the process and ethics of complying with EU ethical testing standards, as well as the ethical test number for informed consent of participants.

2.     All references cited in the article should strictly follow the journal's author's guide.

3.     Please clearly state the four different groups in the abstract.

4.     The manuscript has a lot of all kinds of errors, missing or wrong spaces, awkward wordings. It's hard to read. I have to edit it by two native English speakers and then submit it to other magazines.

5.     All of references should be carefully check out. Some important reference must be added.

6.     From the perspective of scientific discussion, in the discussion part, it is suggested to increase the discussion of persimmon polysaccharide synthesis on animals or cells after exposure of different stimulus sources to animals or cells, so that the discussion part of the article is more complete, and also reveals the intervention effect of persimmon polysaccharide synthesis on intestinal inflammation from different animal models and levels. Through PubMed search, we suggest adding these important literatures to make the discussion part of the article more complete. These important documents including doi:10.1016/j.fct.2023.113863; doi:10.2174/1389200221666200302124102; doi:10.1016/j.fct.2020.111326; doi:10.1155/2020/5363546;  doi: 10.1016/j.envpol.2021.117865. You should support your discussion and results with these recent major publications.

Reviewer 3 Report

It is an appreciable attempt by authors to produce an innovative flour to be used for preparing various processed foods with safe contamination level of DON, suitable for children over 3 years. Although the manuscript contains essential ingredients for the readers but a bit complex to understand. Some other concerns are:  

>> The study lacks control data. Did the authors compare the DON level in the available packaged food products as a control? The evaluation of the transfer of the DON from the raw material to the finished products should be compared with the control.

>> An HPLC analysis was performed in the present study, however the HPLC images are missing. It needs to be added to check the accuracy.  

>>Did the authors detect any other metabolites other than DON? If yes, what was the percentage availability? It is suggested to include results of GC-MS analysis or any other analytical method to confirm the level of DON and to detect other metabolites.

>> Line 357…359.  “The exposure assessment of children consuming the investigated products on a daily basis and the risk characterization by comparing the results of the exposure assessment with the reference toxicological dose”. The results of the consumption of investigated products on a daily basis in children are lacking. Does the study meet the standard least toxicological profiles? How was it assessed?

>> It is mentioned that no studies/ reports available regarding the legal DON exposure in children, so the authors considered adult-based exposure level of DON. However, in conclusion, the authors suggested the products prepared from the innovative flour are suitable/safe for children over 3 years from DON toxicity. In this case, authors may include the data of daily consumption of normal processed food products and DON level in it, consumed by children. 

>> The processing and production cycle of the innovative flour can be represented in a flow chart/ schematic diagram to bring more clarity to the procedure. This can be also useful for the readers to understand it easily. 

The English language is good but the texts should be more organized, concise, and readers-friendly. 

Reviewer 4 Report

Deat authors,

The article is very interesting but I think sometimes is developed in a general way about generally known aspects and does not focus specifically (for example) on the technological treatments that have been used, the references to assess exposure or compare with other previous studies.

Keywords: HACCP should be added instead of Control Plans. Bakery products shoulld be added,

 Introduction

In general, I think the introduction is too broad and focuses the article in a very general sense. Perhaps it should focus more on the technological aspect of producing bakery products and talk less extensively about food safety and legislation.

The technological part and references regarding the consumption data should be included in the introduction. Previous studies regarding DON transfer from flour to bakery products should be added. Other general parts should be omitted because are generally known.

e.g.: “According to the UNICEF Convention on the Rights of the Child [6], approved in 1989, the term child is defined as “every human being below the age of 18 years unless, under the law applicable to the child, majority is attained earlier”.

e.g.: The name of these toxic chemical compounds derives from the Greek μύκης (mushroom) and τοξίνες (toxins)

Line 140 to 142: Please please add a corresponding reference to this affirmation

Results and discussion

I think the heading of the results and the discussion should be in order. The results should be developed first and the exposure assessment should be described at the end.

I also miss in the results and discussion the comparison with other researchers.

Lines 205 to 207: I think you should only talk about food transmission.

Line 249: The Italian consumption data references should be included somewhere, it looks very messy.

Line 267 (table): I would like to know the reference of the body weight (26.1 kg). I have not seen the reference in material and methods or here, I see it messy.

Lines 267 and 268 (tables 1 and 29:  day instead of die. “Kg/bw” instead of “Kg/pc”

Conclusions

Line 270: I see the conclusions as broad and general, I would like to see more targeted conclusions on your results in a more concrete way.

Materials and methods

Nothing is said about the treatment of individual products in material and methods (industry). I think this part should be expanded more and the introduction should be more summarised.

Line 303: μm should me separated from the numbers

Lines 311 and 312: The references of the chromatography equipment and IAC should be added.

Line 337 (table): “Sd” instead of “Sr”.

Line 337 (table): I don't quite understand why the mean coincides with the limit of quantification. Is this an error?

Line 342: I am not clear about the number of samples of each of the products. For example, Are there 6 of bread, focaccia and breadsticks? Biscuits? Plumcake?.

Line 403 and 404: I think it is better to add fp in brackets. For example: in final product corresponds to the contamination of DON in the flour contained in the finished product (fp).

Line 408: The letters in the table are written in different sizes.

Reviewer 5 Report

I expect that there is a publishable body of work here that would be of significant value, but I must suggest that this be reconsidered after major revision.  

First, the authors need to very plainly state their objectives – they make a statement like this in the abstract, but I have no idea what they mean.  I still don’t know how a flour can be ‘innovative.’  

Second, the English / writing is nearly all correct, in the sense of subject / verb agreement, but I don’t understand the meaning.  For example, the last 2 sentences of the abstract seem to argue with each other:  “Documented exposure is well below TDI” and then “We can only have safe foods if we can make flour with much lower DON levels.”  And the repeated use of the word “innovative” is puzzling.  Maybe pick a different word, because I don’t think it means what you think it means.  If I’m wrong, please explain.

I realize that many journals place Materials and Methods at the end of papers, but I would encourage you to consider moving it to the more traditional location, before the results.  In many cases the results are unintelligible without at least a brief reading of the methods.

Line 5-8:  Not really important for this paper, and certainly not essential in the abstract.  Get to the science.

Line 16:  Define TDI in the abstract

Line 18: “Adverse Health effects” is a strangely broad keyword.  “Control plans” is also weird – it could refer to flooding evacuation or fire prevention or a poison hotline.

Line 29:  Why did you capitalize “Good Agricultural Practices”. Is this a title you are referencing?

Line 33-35:  You need to be clear what you’re talking about.  From other parts of your paper I realized you are citing the Official Journal of the European Communities, but it is not clear from this sentence what you’re talking about.

Line 36:  Again, what “Regulation” are you talking about?

Line 63-78:  This is very confusing.  It is easy to see the value of low-mycotoxin food products, but I do NOT see how this is “innovative”.

Line 75:  What is EFSA?

Line 79-116:  This may be a useful justification for your project, but I am struggling to see the how this justifies this particular manuscript.

Line 133:  What is this?  Is this an EU agency?  An Italian Agency?

Line 137:  I don’t disagree with this definition, but how does it effect this manuscript.

Introduction:  This introduction is far too long and written more as a legal brief than a scientific manuscript.  After more than 4 pages the reader still does not know what work is being presented here.  Considering that the entire purpose of this work to to protect children from DON exposure, I would expect much stronger documentation of the dangers of DON.  You only have 2 references to this point and 1 of them rejected DON as a carcinogen.

Line 297-329:  I commend the authors for using robust and widely approved methods for this portion of the work.

Table 3:  I’m confused.  I assume that this is about DON, but “DON” does not appear in the title or any of the headings.  

Table 3:  The methods do not describe how the fortification was performed.  Also, why these levels of fortification?

See above.  The spelling and grammar are fine, but the writing is confusing.  

Round 2

Reviewer 1 Report

Authors must describe the statistical method in details and indicate their findings in tables.

Author Response

Dear Reviewer,

thanks for your remark. As far as the statistics, in consideration of the main target of the paper, what had to be made reliable were the results obtained from the analysis. For this reason, we included the statistics in the procedure of verification of the method of analysis and we consider that this information is sufficient to support all the other experimental findings, since they refer to the content of DON in the various samples analysed. Having said this, we consider that the paper should be accepted in the present form

Reviewer 2 Report

After a round of revisions, I believe that the language of the article should undergo another minor revision. Apart from this, I have no other opinions.

After a round of revisions, I believe that the language of the article should undergo another minor revision. Apart from this, I have no other opinions.

Author Response

Dear Reviewer,

Thanks for your remark. After your first suggestion we dedicated a lot of time for revising the article also for the English form. We consider the present form acceptable from the point of view of the English style as confirmed by the other four reviewers. 

Reviewer 3 Report

The response of most of the questions has been presented logically. So, there is still a scope of improvement!  

It's okay. 

Author Response

Dear Reviewer,

Thanks for your remark. After your first suggestions we dedicated a lot of time to revise the article according the precious comments received. And we all agreed that the final result was much better than the original text. But now, we consider the paper in the present form as ready for sharing with the readers the key message of our findings. 

Reviewer 4 Report

This publication is very interesting and within the scope of the journal.

Best regards

Author Response

Dear Reviewer,

We all would like to thank you for your comments that helped us to ameliorate the content of the paper and the message we obtained from our research. We consider the present form ready to share with the readers the key message of our findings  

Reviewer 5 Report

Much improved!  Thanks to the authors for the patience in cleaning up some problems with the earlier draft.

Line 207:  Bad formatting here.  

Table 3:  Suggest changing column width / text size to clean up the column headings

Author Response

Dear Reviewer,

Without any rethoric, We all would like to thank you for your valuable comments sent about the earlier draft. The corrections in Tables 3 and 4 and the alignment of the text in the row 207 will be inserted in the final version of the MS